# Vacuolal and Peroxisomal Calcium Ion Transporters in Yeasts and Fungi: Key Role in the Translocation of Intermediates in the Biosynthesis of Fungal Metabolites

**DOI:** 10.3390/genes13081450

**Published:** 2022-08-15

**Authors:** Juan F. Martín

**Affiliations:** Departamento de Biología Molecular, Universidad de León, 24071 León, Spain; jf.martin@unileon.es

**Keywords:** calcium transporters, cephalosporin intermediates transport, penicillin biosynthesis, peroxisomes, subcellular compartmentalization, Transient Receptor Potential ion channels (TRP), vacuoles, yeasts/filamentous fungi

## Abstract

**Highlights:**

The intracellular calcium content plays a key role in the expression of genes involved in the biosynthesis and secretion of fungal metabolites.The cytosolic calcium concentration in fungi is maintained by influx through the cell membrane and by release from store organelles.Some MSF transporters, e.g., PenV of *Penicillium chrysogenum* and CefP of *Acremonium chrysogenum* belong to the TRP calcium ion channels.A few of the numerous calcium ion transporters existing in organelles of different filamentous fungi have been characterized at the functional and subcellular localization levels.The cytosolic calcium signal seems to be transduced by the calcitonin/calcineurin cascade controlling the expression of many fungal genes.

**Abstract:**

The intracellular calcium content in fungal cells is influenced by a large number of environmental and nutritional factors. Sharp changes in the cytosolic calcium level act as signals that are decoded by the cell gene expression machinery, resulting in several physiological responses, including differentiation and secondary metabolites biosynthesis. Expression of the three penicillin biosynthetic genes is regulated by calcium ions, but there is still little information on the role of this ion in the translocation of penicillin intermediates between different subcellular compartments. Using advanced information on the transport of calcium in organelles in yeast as a model, this article reviews the recent progress on the transport of calcium in vacuoles and peroxisomes and its relation to the translocation of biosynthetic intermediates in filamentous fungi. The *Penicillium chrysogenum* PenV vacuole transporter and the *Acremonium chrysogenum* CefP peroxisomal transporter belong to the transient receptor potential (TRP) class CSC of calcium ion channels. The PenV transporter plays an important role in providing precursors for the biosynthesis of the tripeptide δ-(-α-aminoadipyl-L-cysteinyl-D-valine), the first intermediate of penicillin biosynthesis in *P. chrysogenum*. Similarly, CefP exerts a key function in the conversion of isopenicillin N to penicillin N in peroxisomes of *A. chrysogenum*. These TRP transporters are different from other TRP ion channels of *Giberella zeae* that belong to the Yvc1 class of yeast TRPs. Recent advances in filamentous fungi indicate that the cytosolic calcium concentration signal is connected to the calcitonin/calcineurin signal transduction cascade that controls the expression of genes involved in the subcellular translocation of intermediates during fungal metabolite biosynthesis. These advances open new possibilities to enhance the expression of important biosynthetic genes in fungi.

## 1. Introduction

Calcium (Ca^2+^) is a divalent ion that plays a key role in cell growth and metabolism regulation in all living beings. Due to its positive charge, this ion might sequester the negatively charged phosphate anion. Both Ca^2+^ and PO_4_^3−^ are key ions that modify proteins and other cellular components [1,2,3]. Calcium binds hundreds of proteins affecting their subcellular localization, their associations and function, which results in changes in cell growth, differentiation and aging. Calcium affects many cellular processes in filamentous fungi, including growth, particularly hyphal tips elongation, morphological differentiation, and production of secondary metabolites (also named specialized metabolites) [4,5,6]. A comprehensive study of the hyphal tip elongation indicate that this process takes place in a step-wise form due to changes in the cytoplasmic calcium concentration [7,8].

Calcium supplementation of 0.25% (22.5 mM) to the penicillin-producing cultures increases the expression of the three penicillin biosynthetic genes *pcb*AB, *pcb*C and *pen*DE by about 40–48% with respect to no supplemented cultures [9] and results in increased production of this antibiotic; this stimulation is affected by calcium complexed with highly phosphorylated phosphopeptides [9,10]. This is one of the more strictly regulated ions in the membrane-surrounded organelles. Some organelles and membrane systems store calcium at high concentrations, including the endoplasmic reticulum, Golgi apparatus and vacuoles. Furthermore, calcium in the peroxisomes plays an important role in signaling transduction during biosynthetic processes [11,12]. Therefore, it is important to investigate the calcium receptor proteins that are involved in calcium translocation out of the storage systems and their role in the transport of intermediates and final products during the biosynthesis of biologically/pharmacologically important secondary metabolites in filamentous fungi. Up to six genes encoding major facilitator superfamily (MFS) type transport proteins have been described as connected to the translocation of β-lactam biosynthetic intermediates in *Penicillium chrysogenum* (syn. *Penicillium rubens*) and *Acremonium chrysogenum*, which has contributed to a better understanding of the penicillin and cephalosporin biosynthesis and secretion [13,14,15,16]. In this article, we describe that some of these MFS proteins are calcium ion transporters and stress-gated transient receptor potential (TRP) channels.

## 2. Role of Calcium in Subcellular Transport Mechanisms

Different nutritional, environmental and stress conditions trigger a calcium-mediated response that affects diverse aspects of fungal metabolism. The Ca^2+^ content in the cytosol, at micromolar concentration, is critically important for the work of cellular biosynthetic mechanisms. Above this concentration, calcium is toxic to cellular metabolism. However, the level of this ion in the store organelles is much higher, in the range of 0.1 to 1 millimolar. The adequate calcium concentration in the cytosol is maintained at a micromolar concentration by: (1) Ca^2+^/H^+^ antiporters, (2) ATP-dependent calcium pumps, and (3) calcium-permeable ion channels in the cytoplasmic membrane, and in the membrane of calcium store organelles. The Ca^2+^/H^+^ antiporters, introduce calcium into either the cells or the store organelles using the proton motive force across the membrane, whereas the calcium ATPases use the energy obtained hydrolyzing ATP to perform the ion translocation. This article is focused on the study of vacuole and peroxisomes calcium transporters since they have been found to play a pivotal role in the transport of precursors and intermediates during the biosynthesis of penicillin [3]. Particular attention is paid to novel developments in calcium transporters involved in the translocation into peroxisomes of cephalosporin biosynthesis intermediates.

## 3. Numerous Transporters Are Involved in the Translocation of Fungal Intermediates of Different Specialized Metabolites

An increasing number of different subcellular localizations of enzymes involved in the biosynthesis of fungal metabolites have been described recently [17]. For example, in *P. chrysogenum* and *Aspergillus nidulans*, the first two enzymes of the penicillin pathway are located in the cytosol, whereas the late enzymes involved in the formation of aromatic penicillins (benzylpenicillin and phenoxymethylpenicillin) are located in peroxisomes [16]. Subcellular localization of the secondary metabolites biosynthetic enzymes is a very dynamic process and requires targeting the adequate cellular localization [17].

The biosynthesis of benzylpenicillin in *P. chrysogenum* is an excellent model for studying the transport of precursors and intermediates through different organelles in filamentous fungi [16,18,19]. Briefly, in this pathway, the amino acids L-α-aminoadipic acid, L-cysteine and L-valine are condensed to form the tripeptide δ(L-α-aminoadipyl)-L-cysteinyl-D-valine (ACV) by a multidomain ACV synthetase (ACVS) encoded by the *pcbAB* gene. The ACV tripeptide is cyclized by the isopenicillin N synthase (IPNS), encoded by the *pcbC* gene forming isopenicillin N (IPN) (Figure 1). In the last step, the L-α-aminoadipyl side chain of IPN is replaced with activated aryl side chains by the IPN acyltransferase (IAT) in a reaction that takes place in peroxisomes [20]. This pathway is controlled by a complex regulatory network of transcription factors in response to environmental and nutritional conditions, including Ca^2+^ concentration [9,21,22]. Electron microscopy studies indicated that the biosynthesis of penicillin is compartmentalized between the cytosol and peroxisomes [23,24] (Figure 1). The two first enzymes in the pathway (ACVS and IPNS) co-localize in the cytosol [25], but the precursor L-α-aminoadipic acid is stored in vacuoles [26] and has to be transported through the vacuole membrane. Finally, the synthesis of aromatic side chain precursors activated as CoA-derivatives by aryl-CoA ligases and the exchange of the L-α-aminoadipyl side chain of IPN by these activated aryl side chains takes place in peroxisomes [27,28,29,30]. Consequently, the addition of agents that increase the intracellular number of vacuoles or traffic vesicles such as 1,3-diaminopropane or spermidine induced penicillin overproduction [31]. The subcellular compartmentalization of intermediates and enzymes in the penicillin pathway requires transport systems to ensure the correct flux of precursors and intermediates.

PenV, a vacuole membrane protein of the Major Facilitator Superfamily, has been reported to play an important role in the formation of the ACV tripeptide, likely supplying amino acids from the vacuole lumen to the cytosolic ACVS [15]. PenM, another MFS transporter, is involved in the translocation of the cytosolic IPN into the peroxisomal matrix [32], where the biosynthesis of penicillin is completed. Transport of phenyl acetic acid into peroxisomes is performed by PaaT (also known as PenT), an MFS drug/H^+^ antiporter with 12 transmembrane spanning domains [14]. This transporter was reported to stimulate penicillin production probably by enhancing the translocation of penicillin precursors across the fungal cellular membranes [33].

### Tranporters Encoded by Genes in the Cephalosporin Gene Cluster

Biosynthesis and molecular genetics of the cephalosporin have been well studied. The biosynthesis of cephalosporin is encoded by two different sets of genes located in chromosomes VII and I [34]. The first gene set encodes enzymes involved in the initial steps of cephalosporin biosynthesis, namely ACV synthetase and isopenicillin N synthase, and also three genes involved in transport systems. Two of these genes, *cefP* and *cefM*, encode peroxisomal or peroxisome-related transport proteins involved in the translocation and secretion of cephalosporin intermediates. In contrast to what occurs in *P. chrysogenum*, the isopenicillin N in *A. chrysogenum* is not converted into benzylpenicillin since *A. chrysogenum* lacks the isopenicillin N acyl transferase [35]; rather, it is converted into penicillin N in an isomerization process that takes place in peroxisomes by the action of the product of two different genes *cefD1* and *cefD2* [36,37,38]; this results in the isomerization of the L-α-aminoadipic acid side chain of isopenicillin N to D-α-aminoadipic. The penicillin N formed, but not the isopenicillin N, is then converted to deacetoxycephalosporin C (DAOC) by the DAOC synthase that expands the five membered thiazolidine ring of penicillin N to the six-membered dehydrothiazine ring of cephalosporin C. The final steps convert DAOC to deacetylcephalosporin C (DAC), which is finally converted into cephalosporin C by an O-acetyltransferase encoded by the *cefG* gene (Figure 2).

The *cefP* gene is adjacent to the genes for the ACVS and IPNS and encodes a 12-TMS protein of the MFS family. Using a fluorescent CefP-DSRED fusion protein, it was demonstrated that CefP is located in the peroxisomal membrane [13]. Mutants disrupted in *cefP* accumulate isopenicillin N but do not produce cephalosporin, indicating that this transporter is implicated in the import of isopenicillin N in the peroxisomes.

A second gene, named *cefM* (for its localization in microbodies), is located within the early cephalosporin gene cluster in chromosome VII and encodes a protein that belongs to family 3 (drug/proton efflux proteins) of MFS transporters. Both the distinct size and the amino acid sequence of CefP and CefM indicates that these transporters play different roles in cephalosporin biosynthesis. Fluorescence microscopy studies evidence that CefM protein is located in small spherical punctuated microbodies [39]. The location of CefP and CefM in the peroxisome membrane and microbodies, respectively, is supported by the presence in both proteins of Pex19 interacting motifs. These motifs are involved in binding protein Pex19 and targeting them to the peroxisomal membrane or peroxisome-related microbody membranes [40]. Disruption of *cefM* completely blocks cephalosporin biosynthesis and results in the accumulation of penicillin N; this intermediate of the pathway in the mutant is not converted into DAC, although the late enzymes of the pathway are functional. This suggests that CefM is involved in the efflux of penicillin N from the peroxisomes to the cytosol.

A third gene, *cefT*, encoding a different 12 MFS transporter is located downstream of *pcbAB*, the gene encoding the ACVS [41]. CefT has 12 transmembrane ^TM^ segments and contains the motifs characteristic of the Drug:H^+^ antiporters. The CefT protein has been located by fluorescent microscopy in cell membranes and vacuoles. Although disruption of *cefT* does not significantly affect cephalosporin C production, the overexpression of this gene results in a 90% increase in antibiotic production [41,42]. Until recently, it was unknown if any of these *Penicillium* or *Acremonium* vacuole or peroxisome transporters were involved in calcium homeostasis. Recently, it has been shown that the vacuolal transporter pen V belongs to the TRP calcium ion transporter class [3].

## 4. Transient Receptor Potential Ion Channels in Fungal Metabolism

The transient receptor potential ion channels (TRP channels) are a subfamily of the calcium ion exchangers. They have been studied in detail in plants [43], mammals, and more recently in yeasts [44], but their role in filamentous fungi is largely unknown so far. TRP ion channels play a variety of physiological roles [45]. In yeasts, they play roles in ion homeostasis, growth, filamentation and pathogenicity and have implications on the biosynthesis of signal molecules and bioactive metabolites [46,47]. These TRPs allow the translocation of calcium from the cytosol to calcium reservoirs by using the proton gradient in the plasma membrane and in membranes of intracellular organelles [48]. The increase in the basal cytoplasmic calcium level is due to a transient opening of the ion channel providing calcium influx from either the extracellular medium or from the calcium store organelles. The model vacuolal transporters Vcx1 and Yvc1 of *Saccharomyces cerevisiae* are described in the next section and compared with the homologous proteins found in *P. chrysogenum*, *A. chrysogenum* and other filamentous fungi. This review does not include studies on the transport of calcium across the cell membrane from the external medium that have been well studied by Kim [49]. In a comparative study with the well-known models of calcium ion exchangers in *S. cerevisiae* and *Candida albicans*, we have found in *P. chrysogenum* a vacuolal calcium ion exchanger and TRP ion channels similar to the yeast proteins Vcx1 and Yvc1, respectively. Similarly, a gene encoding a TRP peroxisomal membrane protein has been found in the cephalosporin gene cluster. In addition, we describe the novel class of TRPs named CSC (for calcium-permeable stress gated cation channel), which includes the PenV protein of *P. chrysogenum* and CefP of *A. chrysogenum*.

## 5. The Model Vcx1 Ion Exchangers of *S. cerevisiae:* Comparison with Homologous Proteins in Filamentous Fungi

In fungal and plant cells, the vacuoles are one of the major calcium store sites, in addition to the endoplasmic reticulum [50,51,52]. Calcium is accumulated inside the vacuoles up to the millimolar concentration by the action of the vacuole membrane Ca^2+^-ATPase [53] and by the influx Ca^2+^ ion exchangers. The best-known influx calcium exchanger in yeasts and filamentous fungi is the yeast Vcx1 [54,55], which belongs to the CAX superfamily of calcium-permeable ion exchangers [56,57,58]. The *S. cerevisiae* Vcx1 protein has a molecular mass of 45 kDa, contains 411 amino acids and is located in the vacuole membrane.

When there is a burst in the cytoplasmic content of calcium, the Vcx1 transporter sequesters the calcium into the vacuoles. This ion transporter has a lower affinity for calcium than the calcium pump ATPase, but it has greater capacity. In addition to calcium, the Vcx1 protein transports Mn^2+^ ions, thus allowing *S. cerevisiae* to grow in high concentrations of either calcium or manganese ions. The energy required for calcium transportation by the Vcx1 exchanger is provided by the proton gradient across the vacuole membrane [59].

Protein members of the Vcx1 family contain 10-11 TM domains clustered in two different groups (six plus five) and are separated by a stretch of acidic amino acids [60]. The TM domains are distributed either (i) homogeneously along the protein with loops of 25 to 35 amino acids between the TM6 and TM7 domains, or (ii) separated into two blocks of six and five TM domains with larger inter-loops of 76 to 240 amino acids between the TM6 and TM7 domains. Topology studies of the Vcx1 location in the vacuole membrane indicate that the N-terminus of the protein is located in the cytosol while the carboxyl end is in the lumen of the vacuole [60,61]; generally, in large Vcx1 homologous proteins, the TM-free N-terminus of the protein is extended. In the Vcx1 protein, loops 1, 3, 7 and 9 are located in the vacuole lumen, while loops six and eight are cytosolic [60].

### 5.1. A family of Multiple Vcx1-like Calcium Transporters in Filamentous Fungi: Characterization of a Protein of P. chrysogenum Homologous to the Vcx1 of S. cerevisiae

The accumulated evidence on the molecular structure and the membrane topology of Vcx1 in *S. cerevisiae* serve to identify similar proteins in filamentous fungi. Of note, in a study of the orthologous proteins of Vcx1 in *P. chrysogenum*, we found five homologs putative “vacuole calcium ion transporters” with 44 to 52% amino acid identity to *S. cerevisiae* Vcx1 (Table 1). The *P. chrysogenum* Vcx1-like proteins will be named hereafter.

PcVcxA to PcVcxE and have orthologous proteins in several filamentous fungi, as shown in Table 1. These proteins range from 440 to 705 amino acids and contain 11 TM domains. These domains are homogeneously distributed in one of these proteins (KZN88098), but in the other proteins, they are separated into two blocks of six and five TMs, showing a different number of amino acids in the loops between the TM6 and TM7 domains. In other transporters, the presence of large intervening sequences between the blocks of TMs has been proposed to play an important role since they contain several motifs that may interact with different regulatory proteins [64]; these large loops may collapse, invaginating into the membrane, forming a channel what will allow the translocation of specific substrates through the organelle membrane [64,65,66]. The presence of several Vcx1 homologs in *P. chrysogenum* suggests that these different homologous proteins might be involved in: (1) recognition by different regulatory proteins that interact with the loops, or (2) binding to distinct substrates that may be transported. None of these five active ion exchanger proteins corresponds to the vacuolal PenV protein or to the peroxisome transporters PaaT or PenM. Proteins homologous to the yeast Vcx1 have similarities in other filamentous fungi (Table 1); Aspergillus species contain genes encoding homologous proteins with about 73% identity to *P. chrysogenum* PcVcxA. Most of these proteins have not been biochemically or topologically characterized except for the XP_012046156 of Cryptococcus neoformans, and the CAX protein (XP_011394995) of N. crassa [65,67].

### 5.2. Characteristics and Localization of the Vcx1-like Transporters of Cryptococcus neoformans and Neurospora crassa

*C. neoformans* produce encapsulated cells [68], which are pathogenic to immunocompromised human hosts. The intracellular calcium level in *C. neoformans* has a profound influence on the virulence of this infection [69,70]. A gene (XP_012046156) of *C. neoformans* encodes a calcium ion exchange protein of 606 amino acids with 48.6% amino acid identity to *P. chrysogenum* PcVcxA. Confocal fluorescent microscopy studies revealed that the *C. neoformans* Vcx protein localizes in the vacuole membrane as also occurs with Vcx1 in *S. cerevisiae*. Deletion of the *C. neoformans vcx1* gene does not affect the cell integrity but reduces the secretion of the glucuronoxylomannan polymer that forms the capsule. Pathogenicity studies in mice indicate that the Vcx1 protein is required for full virulence of *C. neoformans* in vivo. The null *vcx1* mutant becomes hypersensitive to cyclosporine at 35 °C but not at 30 °C, which correlates with the ability of this fungus to infect the human body at 37 °C.

The *vcx1* homologous gene of *N. crassa* encodes a 507 amino acids protein, named CAX, with 50% identity to Vcx1 of *S. cerevisiae* [4,67] and 57% with *P. chrysogenum Pc*VcxA. The location of several calcium ion transporters, including the Vcx1 ortholog in *N. crassa* has been studied using GFP or DS-RED fluorescent-labeled organelle markers and the Vcx1 homologous protein labeled with a different fluorescent marker. The *N. crassa* vacuoles were identified using two different markers, the vacuolal SNARE protein and the A subunit of the vacuolal ATPse. Although the classical large vacuoles were observed in all sections of old hyphae, the vacuole complex also includes small spherical vacuoles and tubular structures identified by these two markers in agreement with previous reports on the heterogeneity of vacuoles [71,72,73]. Experiments using labeled *N. crassa* Vcx1 showed that this protein is located in small spherical organelles and also in large vacuoles [67]. The small spherical vacuoles are concentrated at a hyphal region 50 μm from the hyphal tips, whereas the large vacuoles extend from 500 μm from the hyphae tip to old sections of the mycelium [74]. Both the Vcx1 and the membrane vacuole ATPase marker co-localize in the same two types of vacuoles, confirming that the Vcx1 protein is a real vacuole membrane protein. In summary, in the studied fungi, the Vcx1 protein is a calcium influx transporter driven by a proton gradient across the vacuole membrane to transport calcium, but it does not correspond to any of the biosynthetic intermediate transporters identified so far in *P. chrysogenum*. A search of Vcx1 homologous proteins in the cephalosporin producer *A. chrysogenum* identified a 448-amino-acid protein (KFH45419) with 49.75% identity to the homologous yeast protein, which again does not correspond to any of the biosynthetic intermediate transporters described previously in this fungus.

## 6. Efflux Vacuole Calcium Ion Exchangers That Release Calcium from the Vacuoles to the Cytosol: The Model Yvc1 Transporter in Yeast and Comparison with Filamentous Fungi

Interestingly, although the calcium concentration in the cytosol has to be maintained at a low level, there is a different class of ion transporters that release calcium from the vacuoles in response to a variety of elicitors.

Years ago, a calcium efflux transporter was identified in *S. cerevisiae* and named yeast vacuole calcium conductance ion transporter Yvc1 [75]. This protein contains 675 amino acids, has six TM domains, and is located in the vacuole membrane. Functional studies reveal that this protein is an efflux transporter that translocates calcium from the vacuole back into the cytosol (i.e., it plays the opposite role to the Vcx1 transporter that drives the calcium from the cytosol to the vacuoles). A large part of the calcium content stored in the vacuoles is not easily utilized because it is complexed with polyphosphate. About 10% of the total amount of calcium in the vacuoles can be transported back to the cytosol by the Yvc1 transporter when required [76]. The Yvc1 protein is a calcium/ion transporter of the transient receptor potential (TRP) family of proteins [77].

Characterization of the yeast Yvc1 ion exchanger has provided evidence that this TRP channel responds to several types of stressing signals [6,75,77,78]. These stresses include (1) high osmolarity levels [79,80], (2) physicochemical membrane stretching, (3) changes in the phosphoinositol phosphate concentration [78,81] and (4) oxidative stress [82]. The increment of cytosolic calcium concentration, following its release from the vacuoles, triggers the calmodulin/calcineurin cascade that allows the activation of the transcriptional regulator that mediates the hyperosmotic stress.

A search in *P. chrysogenum* and other fungi reveals that each of them has one single gene encoding a protein orthologous to *S. cerevisiae* Yvc1, with identities ranging between 35 and 40% (Table 1). These proteins normally have seven/eight transmembrane domains, relatively close and distributed in an organization of 5 + 1 + 2 transmembrane segments in contrast to the eleven TMs of the Vcx class. The homologous gene in *P. chrysogenum* encodes the protein ZN84307 of 628 amino acids, which does not correspond to the functionally characterized peroxisomal transporters PenM or PaaT and needs further physiological characterization. One gene of this class, *trp1*, homologous to the yeast *yvc1* has been well characterized in the filamentous fungi *Giberella zeae* (anaform *Fusarium graminearum*) [64]. This gene encodes a protein of 692 amino acids with an identity of 40% to the yeast Yvc1 and 51% to the homologous protein of *P. chrysogenum*.

Heterologous expression of *G. zeae trp1* in a *S. cerevisiae yvc1* null mutant and fluorescent labeling studies showed that this protein localizes in the vacuolal membrane. The *G. zeae* homologous gene responds to several stressing signals, including hyperosmotic stress and oxidative stress by H_2_O_2_ resulting in an increase in cytosolic calcium level. The yeast transformant expressing *G. zeae* TRP1 triggers a sharp rise in the cytosolic calcium level when the cells are submitted to temperature increase (40 °C), suggesting that TRP1 is also sensitive to heat shock. These results clearly indicate that the *yvc1* homologs of filamentous fungi respond to different signals and stressing factors. Ihara et al. [64] proposed that these multiple signals interact with different amino acid sequences located in the cytosolic loops of the TRP1 protein and suggested naming this mechanism “multimodal regulation”.

## 7. A Third Class of Calcium Ion Transporters in Filamentous Fungi Belong to the TRP Family

A new family of TRPs, osmosensitive calcium permeable stress-gated cation Vcx1 ortholog (CSC) transporters, containing a Duf221 domain was identified in *Arabidopsis thaliana*, and later, orthologous genes were found in *S. cerevisiae*, named ScCSC1 [63], and *C. albicans* [44,83]. Expression of the *A. thaliana* and the *S. cerevisiae* genes in hamster ovary cells resulted in high intracellular calcium content, which was induced by high osmolarity in the culture broth. Members of the yeast CSC1 family contain a total of eleven TM domains distributed in two clusters of 3 + 8 in the N-terminal region and in the carboxyterminal moieties.

### 7.1. The P. chrysogenum penV Gene Belong to the TRP-CSC Family

The *P. chrysogenum penV* gene encodes an 832-amino-acid protein, PenV, which is a vacuolal membrane transporter containing a DUF221 domain [15]. This protein has a distribution of the transmembrane domains in two blocks, characteristic of the plant and yeast CSC proteins. Comparative analysis of the *penV* encoded protein with the DUF221 containing *S. cerevisiae* CSC1 indicates that both proteins are related (42% identity) [3]; similar results were obtained when PenV was aligned with the *C. albicans* protein KGQ98295 (913 aa and 46.6 % identity). These results suggest that PenV belongs to the CSC group of proteins that have been fully characterized as calcium-permeable stress-gated ion channel proteins. We have found similar proteins in many *P*. species, *Aspergillus* species and other filamentous fungi of the order Eurotyales, but none of them has been functionally characterized so far. The higher identity to PenV was found in the orthologous proteins of *P. nalgiovense* (98%), *P. roqueforti* (95%) and *P. camemberti* (95%). Different *Aspergillus* species contain similar proteins with identities to PenV ranging from 70 to 80% as *Aspergillus clavatus* (79.5% identity to PenV), *Aspergillus fumigatus* (72%), and lower identity with the distantly related *N. crassa* (930 aa, 45% identity) and *F. graminearum* (866 amino acids, 43%) (Table 1). In filamentous fungi that do not produce α-aminoadipic-acid-derived β-lactams, this transporter may be involved in the translocation of α-aminoadipic acid structural analogs such as glutamic or aspartic acid.

### 7.2. Hybrid Glycogen Debranching Enzymes—Calcium-Permeable Stress-Gated Transporters in Aspergillus Species

Detailed analysis of the calcium-permeable stress-gated transporters (CSC) proteins of different filamentous fungi revealed the presence in some *Aspergillus* species of large proteins that gave a high percentage of identity to PenV, e.g., protein KAF4267877 in *A. fumigatus* CNM-CM8714. These large proteins are glycogen debranching enzymes with a size of 2000 to 2400 amino acids. The large KAF4267877 protein of *A. fumigatus* CNM-CM8714 has two different functional regions; the N-terminal moiety (amino acids 1 to 1595) shows 75% amino acids identity with *P. chrysogenum* glycogen debranching enzyme (KZN90486), whereas the carboxyl half of the protein, annotated as uncharacterized integral membrane protein (amino acids 1596 to 2403), that includes 11 transmembrane domains, shows 74% identity with *P. chrysogenum* PenV (KZN90483). An important question is whether all *Aspergillus* wild-type isolates maintain the same hybrid debranching-CSC1 proteins. Surprisingly, some *Aspergillus* genomes showed two independent genes, one of which encodes the small debranching protein. We found that about one-third of the *Aspergillus* genome entries contain these large glycogen debranching enzymes, whereas two-thirds of them have separated genes encoding free-standing glycogen debranching enzymes of about 1540 amino acids that do not have similarity to PenV. In some other *A. fumigatus* strains, e.g., *A. fumigatus* Af293, the genes XP_749963 orthologous to PenV and XP_749964, encoding a glycogen debranching enzyme (1612 amino acids) are adjacent in the genome. Similar fused glycogen debranching-*penV* genes occur in *A. niger* CBS51388, whereas in *A. niger* ATCC13496, they appear as adjacent independent genes (Figure 3). In *A. nidulans* FGSC A4, there is also a sequence entry (XP_657956) for a fused glycogen debranching-PenV protein of 2378 aa. Interestingly, in the case of *A. niger* CBS51388, the fused gene contains only part of the penV-like gene, and this is compensated by the presence of a full-length penV gene in an adjacent position suggesting that a recombination/duplication has occurred in this region. Since there are several entries in the databases referring to the large debranching-CSC proteins, an error in the sequencing of the encoding genes is unlikely. These results suggest that the large glycogen debranching-PenV enzymes were formed by the fusion of two adjacent genes (Figure 3) encoding functionally distinct enzymes; the probability that the small glycogen debranching encoding genes have been formed by splitting of a large ancestral gene is unlikely since most fungi contain the two separated genes, although we cannot exclude this possibility at the present time.

We have not found these hybrid glycogen debranching enzymes in the genome of fungi other than Aspergillus in the currently available databases. Interestingly, the *P. chrysogenum* PenV lacks similarity to the glycogen debranching enzyme (KZN90486, 1524 amino acids) of this same strain. It is important to know that the genes encoding these two proteins in *P. chrysogenum* are located in the same orientation, separated by about 4 kpb containing genes for two hypothetical proteins (Figure 3).

## 8. Ion Transporters in Peroxisomes of Filamentous Fungi

Peroxisomes have not been traditionally considered to be calcium reservoirs, in contrast to vacuoles, endoplasmic reticulum and mitochondria [12]. The peroxisomal membrane is not easily permeable to large molecules. However, early studies in yeasts and mammalian peroxisomes indicate that there are peroxisomal transporters that facilitate the intercommunication between the cytosol and the peroxisomal matrix [84,85]. Internalization of ions and small molecules appear to have an important role in internal signal reactions in peroxisomes [86].

### The Peroxisomal CefP Transporter Also Belongs to the CSC-TRP Ion Channel Family

In *A. chrysogenum*, order Hypocreales, another β-lactam antibiotic producer [87], there is also a gene of the CSC-TRP class, *cefP*, encoding a protein of 866 amino acids and 42.5% identity to PenV containing a DUF221 motif. The *A. chrysogenum cefP* gene is linked to the early cephalosporin biosynthesis genes *pcbAB* and *pcbC*, and CefP is required for cephalosporin biosynthesis. Interestingly, in contrast to the vacuole membrane localization of PenV, the CefP protein of *A. chrysogenum* was found in the peroxisomal membrane, as shown by fluorescent confocal microscopy studies. The fluorescence of CefP-DS RED and the marker EFG-SKL of peroxisomal proteins fully overlap in *A. chrysogenum* cells [13,32], and the superposition of both fluorescent markers resulted in a yellow fluorescence coincident with the peroxisomes.

This finding suggests that CefP is a calcium-activated stress response ion exchanger that may control the translocation of cephalosporin precursors or the transport of biosynthetic intermediates to peroxisomes in which some of the late cephalosporin biosynthetic steps, involved in the conversion of IPN to penicillin N, take place [36]. Genes similar to *cefP* are conserved in related filamentous fungi of the Hypocreales order, as *Emericellopsis atlantica* (70.5% identity to cefP), *Pochonia chlamydosporia* (63.7%), *Nectria hematococca* (63%) or *G. zeae* (60%), or belonging to the Sordariales order, as *Podospora anserina* (52%) or *N. crassa* (49%) (Table 2). The first two of these fungi contain cephalosporin biosynthesis genes [87,88], whereas the other fungi are not known to contain cephalosporin biosynthesis genes other than *cefP*. Transporters similar to CefP may be involved in the incorporation to peroxisome of intermediates of secondary metabolites different from β-lactams

## 9. Downstream Transduction of the Calcium Level Signal

An important question relates to the molecular mechanism by which the calcium signal mediated by TRPs affects the biosynthesis of penicillin, cephalosporin and other secondary metabolites, i.e., how the cytosolic calcium concentration determines the transport of intermediates and the biosynthesis of the ACV tripeptide, which is one of the major targets of calcium signal in *P. chrysogenum* [14]. There is still little evidence on the molecular mechanism underlying these processes, but initial information suggests that it is mediated by the calmodulin/calcineurin signal cascade. Calcium levels in the cytosol above the normal concentration trigger the calmodulin-calcineurin signal cascade, and this induces expression of the *crc* gene encoding a zinc-finger type regulator that activates transcription of a variety of genes, including those for secondary metabolite biosynthesis [89].

Support for this connection was provided by studies of the Vcx protein of *C. neoformans*, which forms part of a calcium ion channel involved in the biosynthesis of virulence factors; high concentrations of calcium inhibit the growth of the *C. neoformans vcx1* mutants but only in the presence of the signal cascade inhibitor cyclosporine, suggesting that the action of the Vcx1 protein is mediated by calcineurin [65].

Calmodulin is a well-known calcium-binding protein in all eukaryotic cells that serves as a Ca^2+^ sensor and calcium signal transducer [90]. The role of calmodulin in the calcium regulation in the cytosol is well-known [91], but there is little information on the calcium homeostasis and the role of calmodulin in organelles, such as peroxisomes, mitochondria and chloroplasts [12,92,93,94,95]. A probable link of the calcium signal cascade to the transport of secondary metabolite biosynthetic intermediates is that calcitonin/calcineurin controls the transport of intermediates to the adequate subcellular compartment when they are required, but this hypothesis needs experimental confirmation.

## 10. Conclusions and Future Outlook

There are numerous calcium transporters located either in the cell membrane or in organelle membranes in filamentous fungi. The model *vcx1* gene of *S. cerevisiae* and *C. albicans* has been widely studied, but there is limited information on filamentous fungi. There are several homologs of Vcx1 in a given strain of *P. chrysogenum* and some other filamentous fungi. These homologous proteins appear to play different roles in calcium homeostasis. Two other classes of calcium transporters (Yvc1 and CSC) in yeasts and filamentous fungi have been identified as transient receptor potential (TRP) ion channels. Comparative recent studies identified several of these two TRP classes in filamentous fungi, e.g., Ycv1-like protein of *G. zeae* [64]. The molecular function of these calcium TRP channels in relation to the biosynthesis of secondary metabolites, their secretion and differentiation has been elucidated in the last few years, e.g., the *P. chrysogenum penV* gene, identified previously to be involved in the biosynthesis of the tripeptide ACV, a precursor of penicillin and the *A. chrysogenum* CefP, a transporter that incorporates isopenicillin N in peroxisomes, belong to the new TRP class named CSC. The available evidence suggests that TRPs and other calcium transporters of vacuoles, peroxisomes and other organelles play key roles in the translocation of intermediates of the biosynthesis of fungal metabolites that control differentiation and pathogenicity. This research field is still in its infancy, and there are many obscure points regarding the subcellular localization of the numerous membrane transporters. Significant advances are expected in this field by the combination of molecular genetic tools with advanced molecular cytology studies using different fluorescence markers [67], immunoelectron microscopy studies [96], and gold-labelled electron microscopy [28].

In the last decade, evidence has been provided that biosynthesis and secretion of secondary metabolites is an elaborate process that proceeds through several organelles and vesicles (e.g., multivesicular microbodies) [17,97]. Some transporters described in microbodies may be located in peroxisome-related multivesicular structures [39], although the role of these vesicular structures in the secretion of fungal metabolites is unclear [16,98]. Comparative analysis of the calcium ion transporters in yeasts and different filamentous fungi reveals that there are many putative calcium ion transporters in filamentous fungi, but they need to be studied functionally at the molecular level to elucidate their role in fungal biology. This opens interesting possibilities to improve the biosynthesis of fungal metabolites in a directed manner.

## Figures and Tables

**Figure 1 genes-13-01450-f001:**
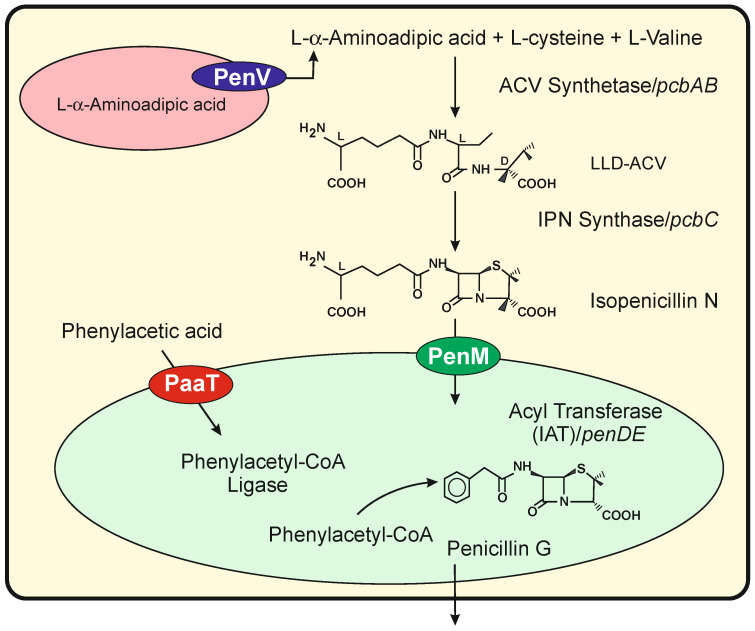
Biosynthetic pathway of benzylpenicillin in *P. chrysogenum*, indicating the subcellular location of the different enzymes and intermediates. A vacuole is shown in pink color, and the PenV transporter in the vacuole membrane is highlighted in a black ellipse. A peroxisome is shown in green color, and the transporters PaaT and PenM are indicated in red and green ellipses, respectively.

**Figure 2 genes-13-01450-f002:**
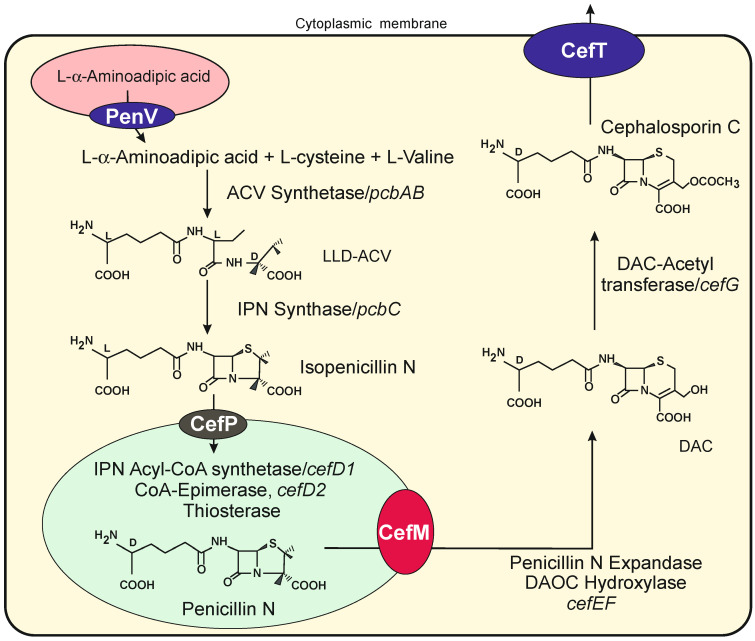
Biosynthetic pathway of cephalosporin C in *A. chrysogenum*, indicating the subcellular location of the different enzymes and intermediates. A vacuole is shown in pink color, and a putative PenV-like transporter in the vacuole membrane is proposed in a black ellipse. A peroxisome is shown in green color, and the tranporters CefP and CefM are indicated in black and red ellipses, respectively. The CefM protein may also be located in multivesicular structures (see text). The transporter CefT is located in the cell membrane.

**Figure 3 genes-13-01450-f003:**
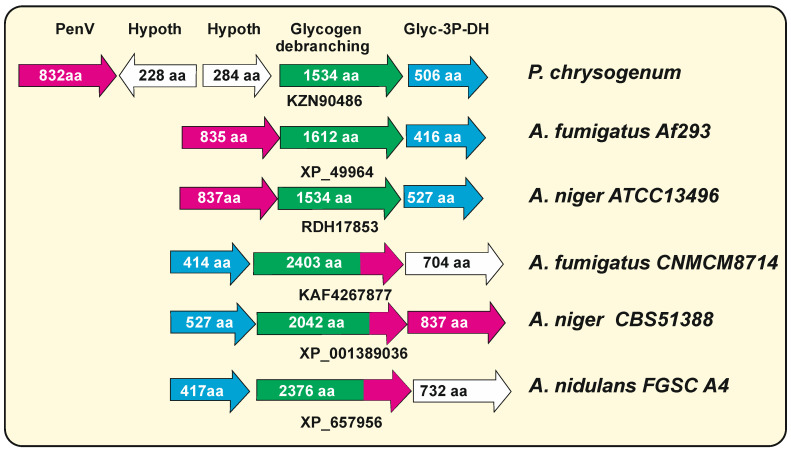
Gene organization of genome region encoding PenV and the glycogen debranching enzymes in different filamentous fungi, listed at the right of the figure. PenV-encoding genes are shown in pink color; genes for glycogen-debranching enzymes are shown in green; genes for glyceraldehyde-3P-dehydrogenase are shown in blue. Hypothetical or unrelated genes are in white. The accession numbers for the respective debranching enzyme are shown below the gene. The amino acid number is indicated inside the arrows. Note that in fused enzymes, the amino terminal moiety is labeled in green, and the carboxy-terminal region is in pink. In *A. fumigatus* CNMCM8714 and *A. nidulans* FGSC A4, the fused protein includes a complete PenV protein (800 and 832 aa, respectively), whereas in *A. niger* CBS52388, the fused protein contains only 504 aa of PenV; in addition, this fungi contains a separate adjacent *penV* gene.

**Table 1 genes-13-01450-t001:** Well-characterized classes of calcium transporters and TRPs in yeast and filamentous fungi in comparison to *S. cerevisiae* proteins ^1,2^.

Microorganism	Influx Transporters Vcx1 Class	Efflux Transporter TRPs Yvc1 Class	Efflux Transporter TRPs CSC Class
*Saccharomyces cerevisiae*	NP_010155, 411 aa, 100%. Vcx1 [60]	QHB11688, 675 aa, 100%. Yvc1 [62]	Q06538, 782 aa, 100 %. CSC. [63]
*Candida albicans*	XP_711893, 416 aa, 62.5%	KHC88137, 675 aa, 44.8%	KGQ98295, 866 aa, 46.8% [44]
*Gibberella zeae*	XP_011319968, 456 aa, 48.9%	TRPGz, XP_011321451. 692 aa, 36.3% [64]	EYB29755, 866 aa, 37.7% XP_011328006, 884 aa, 36.5%
*Penicillium chrysogenum*	KZN93851, 449 aa, 52.5%. VcxA. KZN88098, 440 aa, 45.2%. VcxB. KZN85469, 552 aa, 44.9%. VcxC. KZN88096, 270 aa, 35.3%. VcxD. KZN92764, 461 aa, 34.3%. VcxE	KZN84307, 628 aa, 35.3%	KZN90483, 832 aa, 41.8% [15]
*Cryptococcus neoformans*	OWZ58227, 403 aa, 45,3% OWZ62586, 606 aa, 45,7% [65]	OWZ59838, 622 aa, 30.6%	OWZ78565, 1080 aa, 29.7%
*Neurospora crassa*	XP_011394995, 507, 50%. CAX. [4] AAC08353, 443 aa, 49.5%	KHE84196, 685 aa 37.5%	XP_964945, 930 aa, 37.6%
*Aspergillus fumigatus*	X_750174, 462 aa, 53.55%	XP_001481630, 670 aa, 35.1%	XP_749963, 835 aa, 42.2%
*Acremonium chrysogenum*	KFH45419, 448 aa, 49.75%	KFH43541, 671 aa, 39.7%	KFH48720, 866 aa, 37.6%

^1^ The access number of the proteins, their amino acids (aa) number, the % of amino acids identity to the *S. cerevisiae* model protein and the name of the protein (when it exists) is given. ^2^ In those examples well-documented experimentally, a reference is included. Adapted from Martín and Liras [3] with permission from Elsevier Inc.(Amsterdam, The Netherlands).

**Table 2 genes-13-01450-t002:** CepP-like proteins in different filamentous fungi.

Fungi	Accession Number	Amino Acids	Identity to CefP (%)
*A. chrysogenum*	KFH48720, CefP	866	100.0
*Emericellopsis atlantica*	XP_046122791, CefP	872	70.5
*Tolypocladium paradoxum*	POR38094	875	68.7
*Claviceps lovelessii*	KAH0526729	868	65.2
*Metarhizium anisopliae*	KFG81683	867	65.0
*Fusarium oxysporum*	KAG7415113	858	64.2
*Pochonia chlamydosporia*	XP_018145584	863	63.7
*Trichoderma harzianum*	KKP01098	873	63.2
*Trichoderma virens*	XP_013952278	879	62.6
*G. zeae*	EYB29755	866	61.6
*N. crassa*	XP_964945	930	49.9
*As. fumigatus*	XP_746993	835	43.5
*P. chrysogenum*	KZN90483	832	42.2
*S. cerevisiae*	Q06538 CSC1p	782	36.9
*C. albicans*	KGQ92391	866	34.6
*C. neoformans*	OWZ67604	1080	27.9

The data were obtained from PubMed protein, and the % of identity to CefP was calculated with BLAST.

## Data Availability

All data are available in the manuscript.

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
