# Peer review of "Vacuolal and Peroxisomal Calcium Ion Transporters in Yeasts and Fungi: Key Role in the Translocation of Intermediates in the Biosynthesis of Fungal Metabolites"

_genes, 2022, doi:10.3390/genes13081450_

Round 1

Reviewer 1 Report

Although the transportation of calcium has been widely studied in animal cells acts as an important signal molecule, there is still little information on its role in the biosynthesis of metabolites especially the secondary metabolites on fungi. As said by the author of this manuscript, this field is in its infancy and there are many obscure points such as the localization of the numerous membrane transporters. In this point, this manuscript not only summarizes the progress in this field, but also gives the key points which are worth further studying.

Major comments:

1, As well known, the fermentation media for producing penicillin and cephalosporin C contain CaCO3, whether could CaCO3 play a key role in the production of these secondary metabolites?

2, I would like to know whether the expression of penV or cepP is affected by the concentration of calcium ion?

3, As mentioned in section 6, expressing the G. zeae TRP1 triggers a sharp rise in the cytosolic calcium level when the cells are submitted to temperature increase. Whether it means that increasing the cytosolic calcium level is good for cells to resist heat stress?

Minor comments:

1, The references 39-41 were missing in the main text.

2, Line 261, in vivo should be italic.

3, Line 332, it should be “osmosensitive calcium permeable stress gated cation (CSC) transporters”.

Author Response

Although the transportation of calcium has been widely studied in animal cells acts as an important signal molecule, there is still little information on its role in the biosynthesis of metabolites especially the secondary metabolites on fungi. As said by the author of this manuscript, this field is in its infancy and there are many obscure points such as the localization of the numerous membrane transporters. In this point, this manuscript not only summarizes the progress in this field, but also gives the key points which are worth further studying.

Major comments:

1, As well known, the fermentation media for producing penicillin and cephalosporin C contain CaCO3, whether could CaCO3 play a key role in the production of these secondary metabolites?

Answer: Yes, certainly the calcium concentration in the broth affects significantly the production of penicillin or cephalosporin in P. chrysogenum and A. chrysogenum. We have published previously the biosynthesis of penicillin in defined medium with or without supplementation of calcium (the unsupplemented grows well using calcium in the components and the tap water). Supplementation with calcium increases significantly the production of penicillin (Dominguez-Santos et al. 2017, ref. 9).

2, I would like to know whether the expression of penV or cepP is affected by the concentration of calcium ion?. 

A: Presumably they are affected since in A. chrysogenum these transporter genes form part of the cephalosporin gene cluster; in P. chrysogenum there appears to be co-regulated with the three core genes of the penicillin cluster. However, this point will be confirmed experimentally

3, As mentioned in section 6, expressing the G. zeae TRP1 triggers a sharp rise in the cytosolic calcium level when the cells are submitted to temperature increase. Whether it means that increasing the cytosolic calcium level is good for cells to resist heat stress?

A: The response to heat is clearly shown in the work of Ihara et al (2003, ref. 64) in G zeae. Therefore, should affect the resistance to heat-shock

Minor comments:

1, The references 39-41 were missing in the main text.

A: The references 39 to 41 are now in the final version. Suprisingly when Fig 2 was inserted at that point several text lines were deleted in the final version. I have replaced the paragraph in lines 174-196

2, Line 261, in vivo should be italic. A: Modified in line 289

3, Line 332, it should be “osmosensitive calcium permeable stress gated cation (CSC) transporters”. A: Modified in lines 360

Reviewer 2 Report

The manuscript "Vacuolal and peroxisomal calcium ion transporters in yeasts and fungi: key role in the translocation of intermediates in the biosynthesis of fungal metabolites” (Manuscript ID: genes-1850986) is a review covering the numerous calcium transporters located either in the cell membrane or in organelle membranes in filamentous fungi; calcium transporters that play key roles in the translocation of intermediates of the biosynthesis of fungal metabolites that control differentiation and pathogenicity of these microorganisms. It is a very good work, updated, complete and a great contribution to future work to be undertaken in this area of knowledge.

There are, however, quite a few formatting issues that need to be corrected to improve the document:

Lines 18-19: “tripeptide d-(-a-aminoadipyl-L-cysteinyl-D-valine”, please change to “tripeptide δ-(-α-aminoadipyl-L-cysteinyl-D-valine

Lines 27-29: Keywords are most frequently arranged in alphabetical order

Line 33: “deacetylcephalosporin C” C should not be written in bold

Line 52: “(1,2,3)” Please, separate the numbers to (1, 2, 3), Also applies to lines 56 (5,6) and 494 (17,97)

Lines 55-56: Please add a comma after “morphological differentiation”

Line 59: “22.5mM”. Please, separate the number and the unit: 22.5 mM

Line 72: “of b-lactam”. Please change to “of β-lactam”. This applies also to lines 359, 421 and 441

Line 82: Please add a comma after “Above this concentration”

Line 87: Please add a comma after “across the membrane”

Lines 122,123 and 124: There are three hyphens with spaces that separate a word and are not necessary: diaminopro- pane , compart- mentalization and trans- port

Figures 1, 2 an3. Adjust the image to the center

Line 127: “in P.chrysogenum”. Please, add a space after the dot. Also applies to line 159 “in A.chrysogenum”, and Line 410: “in A.niger CBS52388”

Line 130: Please add “respectively” after “in red and green ellipses”. This applies also in line 162, after “in black and red ellipses”

Line 147: Please add a comma after “what occurs in P. chrysogenum

Line 149: Please change “penicillifn N” to “penicillin N”

Lines 151-152: Please change “L-a-amino- adipic acid” to L-α-amino- adipic acid”

Line 152: Please change “D-a-amino- adipic” to D-α-amino- adipic”. This applies also to lines 357, 359 and 360

Line 155: Please change “dehydrothiazine ring” to “dihydrothiazine ring”

Line 172: Please add a comma after “mammals”

Line 175: Please add a comma after “filamentation and pathogenicity”

Line 186: “C. albicans”. The name of the genus must be written in full the first time it is cited: Candida albicans

Line 204: “calcium pump ATPase. but it has”. Replace the dot with a comma

Line 204: Please add a comma after “In addition to calcium”

Line 208. Please add a dot after “(59)”

Line 212: Please change “separated in two blocks” to “separated into two blocks”. This also applies to line 229.

Line 216: Please, change “(60, 61), and generally,” to “(60, 61); generally,”

Line 217: Please add a comma after “In the Vcx1 protein”

Line 229: Please add a comma after “two blocks of six and five TMs”

Line 232: Please add a comma after “to play an important role”

Line 240: “transporters PaaT or PenM Proteins homologous" Please add a dot after PenM

Line 243: “biochem ically”. Please, delete the extra space between m and i

Line 261. The symbol for Celsius degree is not ºC but °C. This applies also to lines 262. 263 and 324

Line 266: “the Vcx1 ortolog In N”. In starts with an incorrect capital letter.

Line 269: “two different markers the vacuolal SNARE protein and the”. Please, add a colon after markers

Line 279: Please add a comma after “in the studied fungi”

Line 288: Please add a comma after “Interestingly”

Line 291. Please add a comma after “Years ago”

Lines 295-296. “i.e. it plays the opposite role than the Vcx1 transporter that drives the calcium from the cytosol to the vacuoles.”. This text must be placed between parentheses

Line 310. Please add a comma after “transmembrane domains”

Line 313: Please add a comma after “628 amino acids”

Lines 318-319. The two paragraphs are related, remove the full stop. This applies also to lines 430-431 and 455-456

Line 321: Please add a comma after “several stressing signals”

Line 384: Please, avoid unnecessary capital letters “Glycogen”. This applies also to Glycogen again (lines 402 and 404), Databases (lines 387 and 396) and Glyceraldehyde (line 405)

Line 385: Please add a comma after “Interestingly”

Line 399: Please add a comma after “same orientation”

Line 399: Please change kb with kbp

Line 417: “(84, 85 Internal-”. Please add “).” after 85

Line 422: “encoding a of 866 amino acids”. Please add “protein” between a and of

Line 444: Replace Blast with BLAST. It is an acronym.

Line 451: Please add a comma after “underlying this processes”. Also, change “this” with “these”

Lines 456-457. Please add a comma after “neoformans

Line 465. Please add a comma after “94”

Line 490: Please change “and” by a comma

Lines 491. Please add a comma after “(96)”

Author Response

The manuscript "Vacuolal and peroxisomal calcium ion transporters in yeasts and fungi: key role in the translocation of intermediates in the biosynthesis of fungal metabolites” (Manuscript ID: genes-1850986) is a review covering the numerous calcium transporters located either in the cell membrane or in organelle membranes in filamentous fungi; calcium transporters that play key roles in the translocation of intermediates of the biosynthesis of fungal metabolites that control differentiation and pathogenicity of these microorganisms. It is a very good work, updated, complete and a great contribution to future work to be undertaken in this area of knowledge.

There are, however, quite a few formatting issues that need to be corrected to improve the document:

Lines 18-19: “tripeptide d-(-a-aminoadipyl-L-cysteinyl-D-valine”, please change to “tripeptide δ-(-α-aminoadipyl-L-cysteinyl-D-valine. Answer: In the original article sent to the Journal the tripeptide name included the symbols. We have corrected the actual version.

Lines 27-29: Keywords are most frequently arranged in alphabetical order. A: Changed as suggested in lines 28-29

Line 33: “deacetylcephalosporin C” C should not be written in bold. A: Changed as suggested in line 35

Line 52: “(1,2,3)” Please, separate the numbers to (1, 2, 3), Also applies to lines 56 (5,6) and 494 (17,97). A: Changed as suggested in lines 54, 58 and 519

Lines 55-56: Please add a comma after “morphological differentiation” A: included in line 57-58

Line 59: “22.5mM”. Please, separate the number and the unit: 22.5 mM. A: the units have been separated in line 61

Line 72: “of b-lactam”. Please change to “of β-lactam”. This applies also to lines 359, 421 and 441. A: Again, this is an error in the version made by the journal. Has been corrected in lines 74, 385, 446 and 466

Line 82: Please add a comma after “Above this concentration” A: Included in line 84

Line 87: Please add a comma after “across the membrane” A: Included in line 90

Lines 122,123 and 124: There are three hyphens with spaces that separate a word and are not necessary: diaminopro- pane , compart- mentalization and trans- port A: It is an error in the conversion to the actual version. They have been eliminated in lines 124, 125, 126

Figures 1, 2 an3. Adjust the image to the center A: We have tried our best

Line 127: “in P.chrysogenum”. Please, add a space after the dot. Also applies to line 159 “in A.chrysogenum”, and Line 410: “in A.niger CBS52388”. A: Changed in lines 139, 168, 408, 411. The changes in the lines are due to the modified position of the figures

Line 130: Please add “respectively” after “in red and green ellipses”. This applies also in line 162, after “in black and red ellipses”. A: The word “respectively” has been added in lines 142 and 171

Line 147: Please add a comma after “what occurs in P. chrysogenum” A: Included in line 150

Line 149: Please change “penicillifn N” to “penicillin N” A: corrected in line 152

Lines 151-152: Please change “L-a-amino- adipic acid” to “L-α-amino- adipic acid” A: corrected in 154-155

Line 152: Please change “D-a-amino- adipic” to “D-α-amino- adipic”. This applies also to lines 357, 359 and 360. A: Corrected in lines 154-155, 385 and 386

Line 155: Please change “dehydrothiazine ring” to “dihydrothiazine ring” A: Corrected in line 155

Line 172: Please add a comma after “mammals” A: included in line 199

Line 175: Please add a comma after “filamentation and pathogenicity” A: included in line 202

Line 186: “C. albicans”. The name of the genus must be written in full the first time it is cited: Candida albicans. A: You are right. It has been changed in line 213

Line 204: “calcium pump ATPase. but it has”. Replace the dot with a comma A: Corrected in line 232

Line 204: Please add a comma after “In addition to calcium” A: Corrected in line 232

Line 208. Please add a dot after “(59)” A: Corrected in line 236

Line 212: Please change “separated in two blocks” to “separated into two blocks”. This also applies to line 229. A: Corrected in lines 240 and 257

Line 216: Please, change “(60, 61), and generally,” to “(60, 61);,” A: Corrected in line  244

Line 217: Please add a comma after “In the Vcx1 protein” A: Corrected in line 245

Line 229: Please add a comma after “two blocks of six and five TMs” A: Corrected in line 257

Line 232: Please add a comma after “to play an important role” A: Corrected in line 260

Line 240: “transporters PaaT or PenM Proteins homologous" Please add a dot after PenM A: Corrected in line 273

Line 243: “biochem ically”. Please, delete the extra space between m and I A: Corrected in line 276

Line 261. The symbol for Celsius degree is not ºC but °C. This applies also to lines 262. 263 and 324. A: The symbol has been corrected in lines 289, 290, 291 and 352  

Line 266: “the Vcx1 ortolog In N”. In starts with an incorrect capital letter. A: Corrected in line 294

Line 269: “two different markers the vacuolal SNARE protein and the”. Please, add a colon after markers A: Corrected in line 297

Line 279: Please add a comma after “in the studied fungi” A: Corrected in line 307

Line 288: Please add a comma after “Interestingly” A: Corrected in line 316

Line 291. Please add a comma after “Years ago” A: Corrected in line 319

Lines 295-296. “i.e. it plays the opposite role than the Vcx1 transporter that drives the calcium from the cytosol to the vacuoles.”. This text must be placed between parentheses A: Corrected in line 323-324

Line 310. Please add a comma after “transmembrane domains” A: Corrected in line 338

Line 313: Please add a comma after “628 amino acids” A: Corrected in line 341

Lines 318-319. The two paragraphs are related, remove the full stop. This applies also to lines 430-431 and 455-456 A: Corrected in lines 373-374, 454, 483

Line 321: Please add a comma after “several stressing signals” A: Corrected in line 349

Line 384: Please, avoid unnecessary capital letters “Glycogen”. This applies also to Glycogen again (lines 402 and 404), Databases (lines 387 and 396) and Glyceraldehyde (line 405). A: Corrected in lines 408, 423, 424 for glycogen, 414 and 432 for database and 425 for glyceraldehyde

Line 385: Please add a comma after “Interestingly”  A: Corrected in line 411

Line 399: Please add a comma after “same orientation” A: Corrected in line 435

Line 399: Please change kb with kbp”  A: Corrected in line 435

Line 417: “(84, 85 Internal-”. Please add “).” after 85  ”  A: Corrected in line 442

Line 422: “encoding a of 866 amino acids”. Please add “protein” between a and of  ”  A: Corrected in line 447

Line 444: Replace Blast with BLAST. It is an acronym”  A: Corrected in line 469

Line 451: Please add a comma after “underlying this processes”. Also, change “this” with “these” A: Corrected in line 476

Lines 456-457. Please add a comma after “neoformansA: Corrected in line 482

Line 465. Please add a comma after “94” A: Corrected in line 490

Line 490: Please change “and” by a comma A: Corrected in line 515

Lines 491. Please add a comma after “(96)”  A: Corrected in line 515